# Performance Evaluation of Vibrational Measurements through mmWave Automotive Radars †

**Gianluca Ciattaglia** [1], **Adelmo De Santis** [1], **Deivis Disha** [1], **Susanna Spinsante** [1], **Paolo Castellini** [2] and **Ennio Gambi** [1,*]

[1] Information Engineering Department, Università Politecnica delle Marche, via Brecce Bianche 12, 60131 Ancona, Italy; g.ciattaglia@pm.univpm.it (G.C.); adelmo.desantis@univpm.it (A.D.S.); d.disha@pm.univpm.it (D.D.); s.spinsante@univpm.it (S.S.)

[2] Department of Industrial Engineering and Mathematical Sciences, Università Politecnica delle Marche, via Brecce Bianche 12, 60131 Ancona, Italy; p.castellini@univpm.it

\* Correspondence: e.gambi@univpm.it

† This paper is an extended version of our paper published in Proceedings of 2020 IEEE 7th International Workshop on Metrology for AeroSpace (MetroAeroSpace), Pisa, Italy, 22–24 June 2020.

**Abstract:** Thanks to the availability of a significant amount of inexpensive commercial Frequency Modulated Continuous Wave Radar sensors, designed primarily for the automotive domain, it is interesting to understand if they can be used in alternative applications. It is well known that with a radar system it is possible to identify the micro-Doppler feature of a target, to detect the nature of the target itself (what the target is) or how it is vibrating. In fact, thanks to their high transmission frequency, large bandwidth and very short chirp signals, radars designed for automotive applications are able to provide sub-millimeter resolution and a large detection bandwidth, to the point that it is here proposed to exploit them in the vibrational analysis of a target. The aim is to evaluate what information on the vibrations can be extracted, and what are the performance obtainable. In the present work, the use of a commercial Frequency Modulated Continuous Wave radar is described, and the performances achieved in terms of displacement and vibration frequency measurement of the target are compared with the measurement results obtained through a laser vibrometer, considered as the reference instrument. The attained experimental results show that the radar under test and the reference laser vibrometer achieve comparable outcomes, even in a cluttered scenario.

**Keywords:** mmWave; FMCW; radars; automotive; vibrations; micro-Doppler



## 1. Introduction

Moving from our previous paper on the topic [1], a more accurate description of the work carried out is presented in this manuscript, where the theoretical background has been extended and details about the measurements have been more deeply analyzed and explained.

Radar systems play a very important role in modern cars since many years, before being nowadays exploited in autonomous driving vehicles. Radars can be used for different applications, as an example, for door opening warning, or park assistance to the driver. For each application, specific norms regulate the type of radar that it is possible to use [2,3]. During the time, the operating frequencies have been moved from the initial 24 GHz band to the 77 GHz one, and the reason of this shift is the increased necessity of larger bandwidth from other communication systems, such as the 5G [4]. As a consequence, radar manufacturers have moved their systems between these two frequency bands [5]. In most of the cases, these radars use the FMCW modulation technique, because of its simplicity (compared to other modulation techniques) and the use of fewer hardware components inside the devices, which reduces costs and complexity. FMCW Radar sensors are at present widely available in the market, since, thanks to technology advances, their cost

and form factor have been shrinking over time. Many of these Radar sensors are designed for automotive and/or industrial applications, with working frequencies in the mmWave bands. In typical applications, FMCW Radars are used to detect objects inside an area [6,7], making it possible to detect the distance of the targets from the sensor, their radial velocity and their angle of arrival. Regardless of the application in the automotive field, these systems can be classified on the basis of distance sensing capability. In Table 1 the classification is reported [8].

**Table 1.** Classification of automotive radars based on range measurement capability.

| Radar Type | Long-Range Radars | Medium-Range Radars | Short-Range Radars |
|---|---|---|---|
| Range [m] | 10–250 | 1–100 | 0.15–30 |
| Azimuthal feld of view [deg.] | ±15 | ±40 | ±80 |
| Elevation feld of view [deg.] | ±5 | ±5 | ±10 |
| Sample Applications | Automotive cruise control | Lane change assist | Park assist |

The range capability of the sensor depends not only on the transmitted power and the antenna design, but on the sampling frequency of the ADC inside the device. With these Radars, the ADC can sample the signal with a lower frequency than the carrier frequency. On one hand, this is an advantage because it allows to use cheaper converters, but on the other hand it is also a drawback, because the maximum detectable range is related to the beat frequency, and the maximum value will correspond to the maximum sampling frequency of the ADC. In general, an FMCW Radar transmits chirps, that are reflected from the targets inside the observed scene and, by a multiplication between these signals, it is possible to extract the range information. In Figure 1 a basic scheme of the FMCW Radar sensor is given.

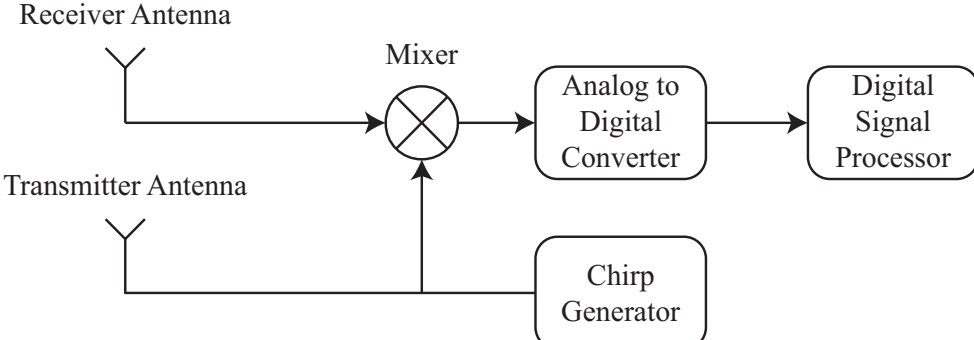

**Figure 1.** Basic scheme of FMCW Radars.

In general, in a Radar system, the range estimation derives from the time delay between the transmitted signal and the returning echo signal. So, it is possible to evaluate the range based on the following equation [9]:

$$R = \frac{c \cdot \tau}{2} \tag{1}$$

where $c$ is the speed of light and $\tau$ the echo delay. In an FMCW Radar, the range estimation derives directly from the product between the transmitted and the received signals. The result of this product is a signal called Beat signal and its frequency depends on the echo delay. With this type of Radars it is also possible to estimate the velocity of the targets in the scene: again, this is possible by evaluating the frequency of the Beat signal, according to:

$$f_b = \frac{2R\Delta_f}{cT} + f_d \tag{2}$$

were $f_b$ is the frequency of the beat signal, $R$ is the distance of the target from the Radar, $\Delta_f$ is the frequency difference between the transmitted and the received chirp, and $T$ is the chirp time. Another term appears in the equation, namely $f_d$, which represents the value of the Doppler effect caused by a moving target [10]. A graphical representation of the transmitted and received signal chirps is depicted in Figure 2.

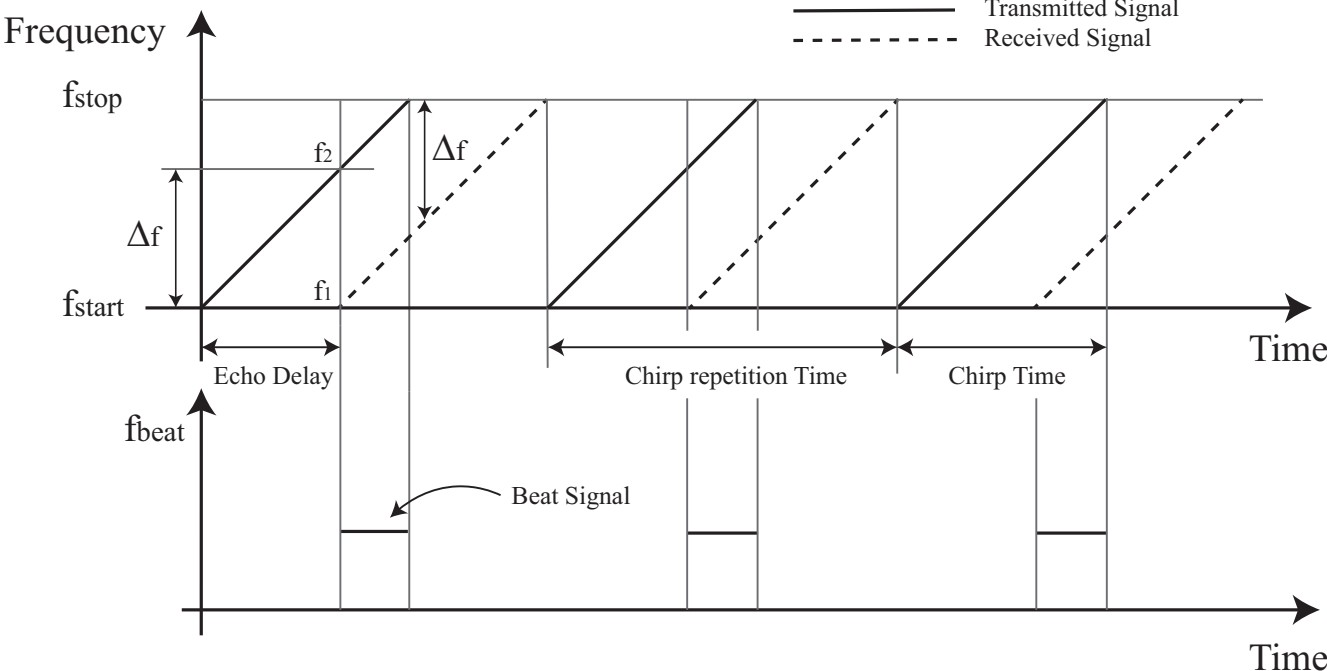

**Figure 2.** Transmitted and received chirps in a Radar system.

To estimate the velocity of the target, the transmission of multiple chirps is necessary. The performances of the system are related mainly to these parameters: the bandwidth used in each chirp, the chirp repetition time, the slope of the chirps, and also the wavelength. From [11], it is possible to estimate the range resolution $\Delta_R$ and the velocity resolution $\Delta_v$ based on the following equations:

$$\Delta_R = \frac{c}{2\Delta_f} \tag{3}$$

$$\Delta_v = \frac{\lambda}{2T} \tag{4}$$

Another important feature used in these systems is the MIMO technology. The estimation of the angle of arrival of the targets is necessary for many ADAS applications, but the problem relies in the size of the Radar device, which must be very small to allow its integration in the vehicle. For this reason, many automotive radars use the MIMO technique to improve their angular resolution. This technique is well-known in communication systems but it can be also exploited for improving the angular resolution of Radar systems. The basic idea is to measure the phase difference between signals at a receiver array. This is possible by performing a FFT along the spatial axis, using the signals' samples. In Figure 3 this principle is depicted by using two receiver antennas and two transmitters. It is important to specify that the target must be in far-field so the paths travelled by the reflected signals can be considered as parallel.

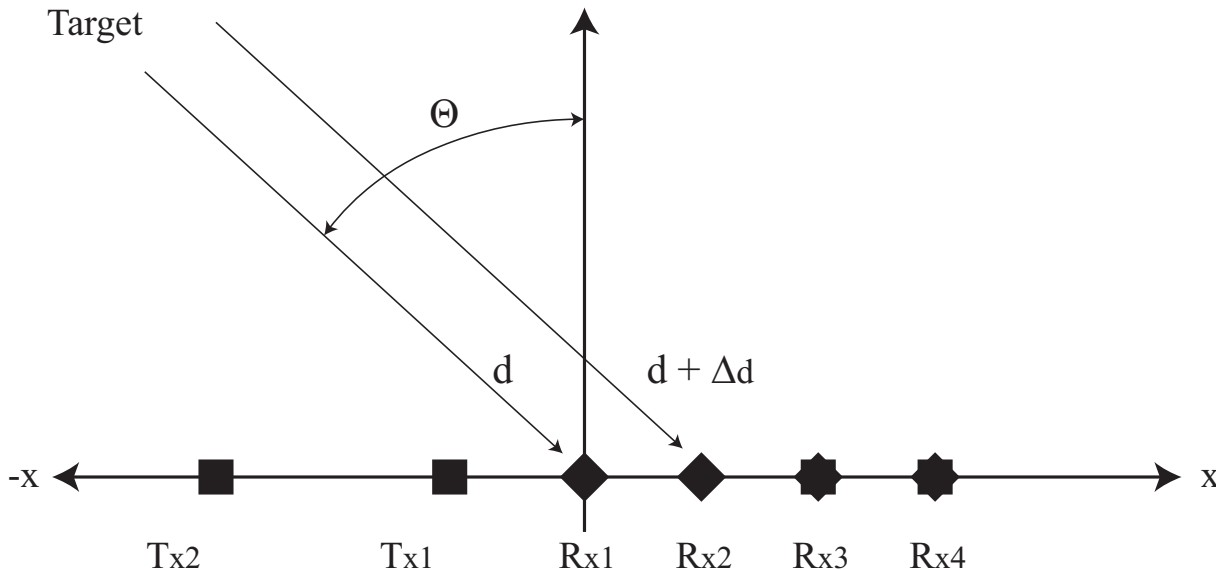

**Figure 3.** Generic MIMO antenna of an automotive Radar system. The squares are the transmitters, the rhombuses the receivers, and the stars the virtual elements of the array.

In Figure 3 only the transmitters and the Rx1 and Rx2 receivers are real elements of the antenna; Rx3 and Rx4 are virtual elements. The phase difference $\Delta_\phi$ at each receiver antenna depends on the distance $\Delta_d$ between the elements, and can be calculated from:

$$\Delta_\phi = \frac{2\pi\Delta_d}{\lambda} \tag{5}$$

The $\Delta_\phi$ changes if the transmitter is Tx1 or Tx2, so, considering the receiver elements spaced by a distance of $\lambda/2$, and Tx2 at a distance of $\lambda$ from Tx1, the $\Delta_\phi$ at Rx1 is 0 when Tx1 transmits, and $2\Delta_\phi$ when Tx2 transmits. The same happens for Rx2, where the phase difference is $\Delta_\phi$ when Tx1 transmits, and $3\Delta_\phi$ when Tx2 transmits. For this reason it is possible to extend the MIMO array with two additional elements, and the new array is called Virtual Array. The elements Rx3 and Rx4 are virtual and generated by the phase delay. This way, the angular resolution can be improved, but still keeping a physically smaller antenna.

As described in [12], for a target with an angle of arrival equal to $\Theta$, the reflected signal arriving at the receivers has a spatial frequency equal to:

$$\omega_1 = \frac{2\pi d}{\lambda} \cdot sin(\Theta) \tag{6}$$

A second target can have the same spatial frequency, but the sine argument is $(\Theta + \Delta_\Theta)$ so the difference between these two spatial frequencies can be calculated as:

$$\Delta_\omega = \omega_1 - \omega_2 = \frac{2\pi d}{\lambda}\left(sin(\Theta + \Delta_\Theta) - sin(\Theta)\right) = \frac{2\pi d}{\lambda}\left[cos(\Theta) \cdot \Delta_\Theta\right] \tag{7}$$

These two spatial frequencies are separated by $\Delta_\Theta$ and this value is visible inside the FFT computation. The number of points used in the FFT depends on the number of virtual elements. The Radar condition to resolve two targets in the angular domain is:

$$\Delta_\omega > \frac{2\pi}{N} \Rightarrow \frac{2\pi d}{\lambda}\left(cos(\Theta) \cdot \Delta_\Theta\right) > \frac{2\pi}{N} \Rightarrow \Delta_\Theta > \frac{\lambda}{Nd \cdot cos(\Theta)} \tag{8}$$

where $N$ is the number of virtual elements of the receiver antenna. In general, the receiver antenna elements are spaced by a $\lambda/2$ distance so, at the end, the angular resolution for an FMCW MIMO radar is:

$$\Theta_{res} = \frac{2}{N} \tag{9}$$

By observing Equation (9) it is possible to understand why the use of the MIMO technique is important. In order to attain better angular resolution, the device needs more receiver elements, but this is a drawback in terms of physical dimensions of the device. By implementing MIMO, it is possible to improve the virtual elements of the array while maintaining a small form factor at the same time.

### 1.1. Radar micro-Doppler Effect for a Vibrating Target

With the exploitation of the Doppler effect it is possible to measure the velocity of the targets, but there is another interesting effect in Radar systems. Suppose that our target moves in the direction of the Radar system, towards it: this movement produces a Doppler effect depending on the relative velocity between Radar and target. It is possible also that the target is composed of any multiple parts, that could be rotating or moving independently from the bulk. For example, in the case of a vehicle, the rotating elements can be the wheels; in the case of a moving person, arms or legs can be the moving elements [13]. These movements produce another Doppler effect called micro-Doppler and it is possible to exploit this effect to identify the target signature, or to measure other parameters [14,15]. By extracting the micro-Doppler information from a Radar signal, it is possible to extend the application field of such devices and use them to detect small vibrations [16,17]. In fact, the vibration of the target generates a phase modulation in the received Radar signal, and by applying specific algorithms, it is possible to extract the phase information [18,19]. To explain how the micro-Doppler appears in the signal generated by a vibrating target, it is possible to resort to the discussion and the Radar model presented in [20,21]. The latter is shown in Figure 4.

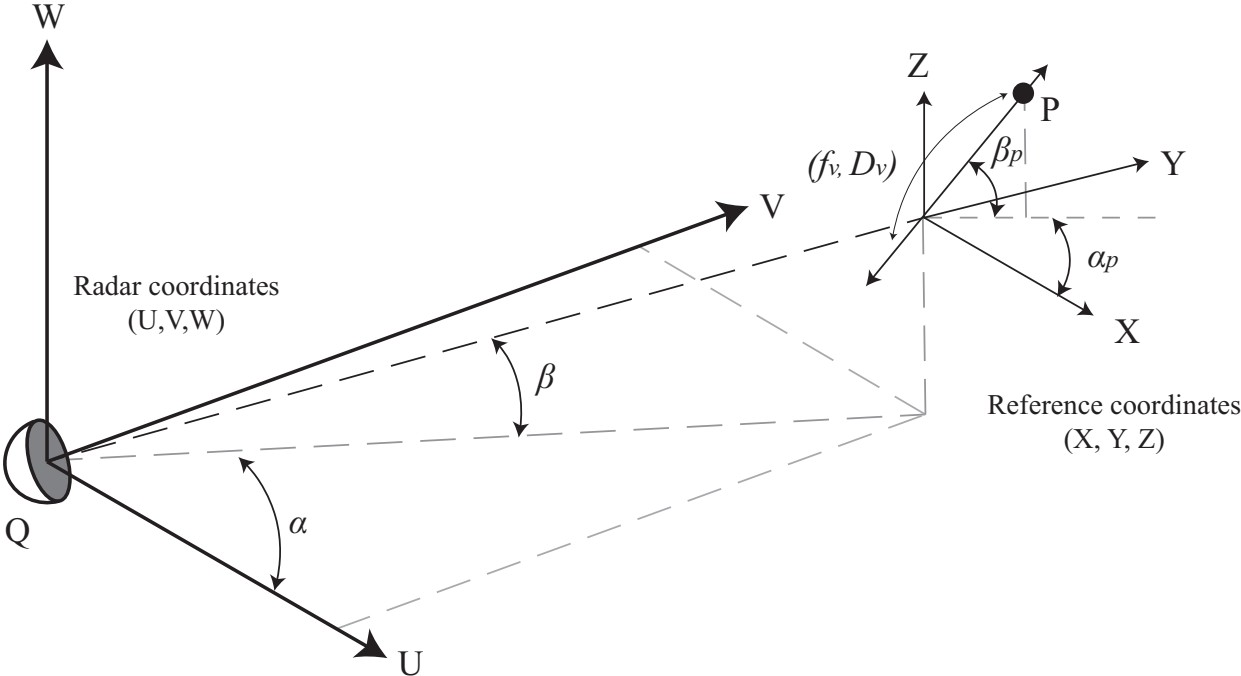

**Figure 4.** Geometry for the Radar and a vibrating point target. The Radar is situated in position Q, and the target is in P.

It is important to understand how in general the micro-Doppler effect works, and its mathematical model can be derived from the classical Doppler formulation. The target located in position P can be vibrating: in this case, its movement can be not only along

the radial axis of the Radar system. The Radar can detect vibrations that are along a radial direction so, for the sake of simplicity, we suppose that the target vibrates in this direction. It is possible to identify the movement of the target by using two parameters: the main vibration frequency $f_v$, and the displacement $D_v$. In general, the frequency is not a single tone but, for the time being, it is possible to use a simplified model. Under these assumptions, it is possible to represent the vibration model of the target as:

$$D(t) = D_v \cdot sin(2\pi f_v t) \cdot cos(\beta) \cdot cos(\alpha_p) = D_v \cdot sin(2\pi f_v t) \qquad (10)$$

where $\beta$ and $\alpha_p$ are taken equal to zero to simplify the model. These angles represent, respectively, the elevation angle of the Radar system with respect to the plane (U,V), and $\alpha_p$ is the radial direction of the displacement. If we denote by $R_0$ the distance between the Radar and the target, the range distance can vary according to the law $R(t) = R_0 + D(t)$. At this point, the received signal becomes:

$$s(t) = \rho \cdot exp\left[j\left(2\pi f_0 t + 4\pi \frac{R(t)}{\lambda}\right)\right] = \rho \cdot exp\left[j(2\pi f_0 t + \Psi(t)\right] \qquad (11)$$

where $\rho$ is the backscattering coefficient, $f_0$ is the carrier frequency and $\lambda$ is the wavelength. As defined above, $R(t)$ is the displacement of the target so it is possible to substitute these terms inside Equation (11), and the result is a new formulation of $s(t)$:

$$s(t) = \rho \cdot exp\left[j\left(4\pi \frac{R_0}{\lambda}\right)\right] \cdot exp\left[j2\pi f_0 t + D_v sin(2\pi f_v t) \cdot \frac{4\pi}{\lambda}\right] \qquad (12)$$

The equation is composed by two terms, the former is related to the main distance of the target $R_0$, the latter is related to the vibrational movement of the target. It is now easy to extract the phase of this signal and evaluate the micro-Doppler part of the total Doppler effect. By indicating the quantity $2\pi f_v$ with $\omega_v$, it is possible to re-write the micro-Doppler expression as:

$$f_m D = \frac{\omega_v D_v}{\pi \lambda} \cdot cos(\omega_v t) \qquad (13)$$

As specified above, this model doesn't take into account the direction of the vibrations. In a more general model, $\beta$ and $\alpha_p$ are not zero and the equation must include also these terms. But for the aims of this work it is enough to understand the principle of the vibration analysis performed with a Radar system. At this point it is possible also to understand how the wavelength plays a key role in the detection of small vibrations. This parameter identifies the vibration resolution capability of the system: the higher the carrier frequency, the better the resolution. For example, by using a mmWave Radar it is possible to detect vibrations with a displacement in the order of microns. Another remarkable feature of this technique is the use of MIMO: thanks to the angular detection capability, the Radar system can simultaneously detect many targets, so it is possible to measure the vibration of any object inside the Radar scene. This is not possible when using a not MIMO radar and the vibrations are summed together for a certain direction. In Section 2, the algorithm to extract the vibration by using an automotive radar is illustrated.

### 1.2. Related Work

The usage of a Radar system for detecting small vibrations is possible with different Radar technologies. As described above, the micro-Doppler effect is the same in all Radar systems, but the signal processing techniques to extract the information on it can be different.

In an FMCW Radar, the transmitted signal is a chirp, so the value of $f_0$ changes along the time. In this modulation scheme, the micro-Doppler effect is extracted from the Beat signal, as previosuly shown, and in [22,23] different approaches are proposed. The detection of an object vibration is possible not only with a terrestrial Radar system, but also with aerospace-born Radars. In [24,25], different techniques using SAR are illustrated. In this

work, the focus is on automotive Radars, so the discussion will be limited to this type of sensors.

Exploiting the information about the vibration-related micro-Doppler in a Radar system can extend the range of its possible applications. There are many examples of the application of these sensors, not only for position detection but also for medical or industrial purposes. It is possible, for example, to detect the vital parameters of several patients inside a clinical setting [26]. The human heart beat can be detected because the heart can be seen as a vibrating object that produces a micro-Doppler signature in the Radar echo. Respiration as well, even if with a simpler detection methodology, produces the same effect and can be detected through micro-Doppler [27–29]. The use of a Radar for such purposes allows to implement a contactless measurement technique, which can be very powerful when the patients' conditions are not compatible with the application of contact sensors.

The same principle can be also applied to monitor the structural health of buildings or bridges [30–32]. Other industrial applications include the detection of water levels inside a tank [33], or the estimation of tyre wear [34].

The transmitted signal wavelength plays a key role in the Radar sensor ability to detect small vibrations, and determines its vibrational resolution [35]. Radars operating at frequencies up to 60 GHz and beyond are classified as mmWave on behalf of the wavelength of the emitted signal. Such a range of wavelength values enables sensors to detect a vibration in the range of hundreds of microns, which is a great result, considering the relatively low cost of the device. Moreover, this technique allows for a high sampling rate, that makes it possible to analyze a wide spectrum of vibrational frequencies, thus making Radar devices even more versatile. Comparing, for example, the results of [36] with [37], it is possible to assert that using a higher frequency makes it possible to detect smaller vibrations. This is possible also with a lower frequency, but the system will be more complex and more expensive. Thanks to the growth of the automotive market, driving the manufacturers of these devices, automotive Radar sensors are easily available for this type of applications, and, as described above, these Radars work with FMCW modulation. Such a modulation has an advantage against the CW technique: if many targets are present in the measurement area, the CW technique does not make it possible to discriminate them along the range, and the detected vibrational signal will be not accurate. CW Radars are not able to detect multiple target vibrations and they need a more cleared measurement area. For this reason, CW Radars are well suited for the detection of vital signs [38,39], but for more general-purpose applications, FMCW Radars provide better performances.

In this work we aim to evaluate the accuracy attainable from an FMCW Radar sensor, in terms of vibrational analysis in a real scenario, in which measurements are carried out inside a mechanical measurements laboratory at Università Politecnica delle Marche. The paper is organized as follows. Section 2 presents the materials and methods used in the evaluation of the Radar sensor performance and its comparison to a reference laser vibrometer. Section 3 describes the experiments performed, the results of which are discussed in Section 4. Finally, Section 5 concludes the paper.

## 2. Materials and Methods

### 2.1. Materials

Two measurement systems are compared in this work, namely a Radar system, consisting of a Radio Frequency (RF) board and a support board, and a Polytec Scanning Vibrometer equipment assumed as the reference instrumentation [40]. It is possible to use different methodologies for the comparison of the two measurement systems: in [41], a laser vibrometer is used, but it is possible to use even some accelerometers [42,43]. In the case of the proposed setup, it is difficult to apply the accelerometers so the laser vibrometer has been chosen as the best system to compare with. The used device is an AWR 1642 from Texas Instruments [44]. It is an FMCW Radar with MIMO capability, as it is equipped with 2 transmitters and 4 receivers. By using MIMO technology we can determine the target

angular position without moving the sensor. The angular resolution is directly bound to the number of transmitters and receivers used [45]. Another remarkable feature of this Radar is its bandwidth, up to 4 GHz. For example, using all the available bandwidth capability, the device gives a range resolution of 0.0375 m. This value results from:

$$R_{resolution} = \frac{c}{2 \cdot B} \tag{14}$$

where $c$ is the speed of light and $B$ is the radar bandwidth. The Radar sensor is an integrated system which encompasses the RF section, a DSP and a controller, managing the entire board. Signals coming from the antenna array are pre-processed in the board and for this reason it is impossible to access the raw collected data. By using a FPGA equipped daughterboard, called DCA 1000 EVM [46], we can gain access to raw samples extracted from the Beat signal too. The daughterboard connects to the Radar using a LVDS bus, and to a computer using an Ethernet connection and the User Datagram Protocol (UDP). This way, we can process raw data on a workstation to analyze them and extract the relevant information.

The Radar transmits a series of chirps: the start frequency is 77 GHz and the stop frequency depends on the Radar configuration but can reach 81 GHz as a maximum. Figure 5 shows how the sensor generates the chirps. The description of the parameters follows:

- Idle Time: time required for the ramp generator to return to its original state;
- ADC Valid Start Time: idle time used to remove data in the very beginning of the ramp. This way we can improve system linearity and reduce distortion in the beat signal;
- ADC Sampling Time: amount of time during which ADC samples the beat signal;
- Used Radar Bandwidth: effective Radar bandwidth after the initial part of the ramp has been removed.

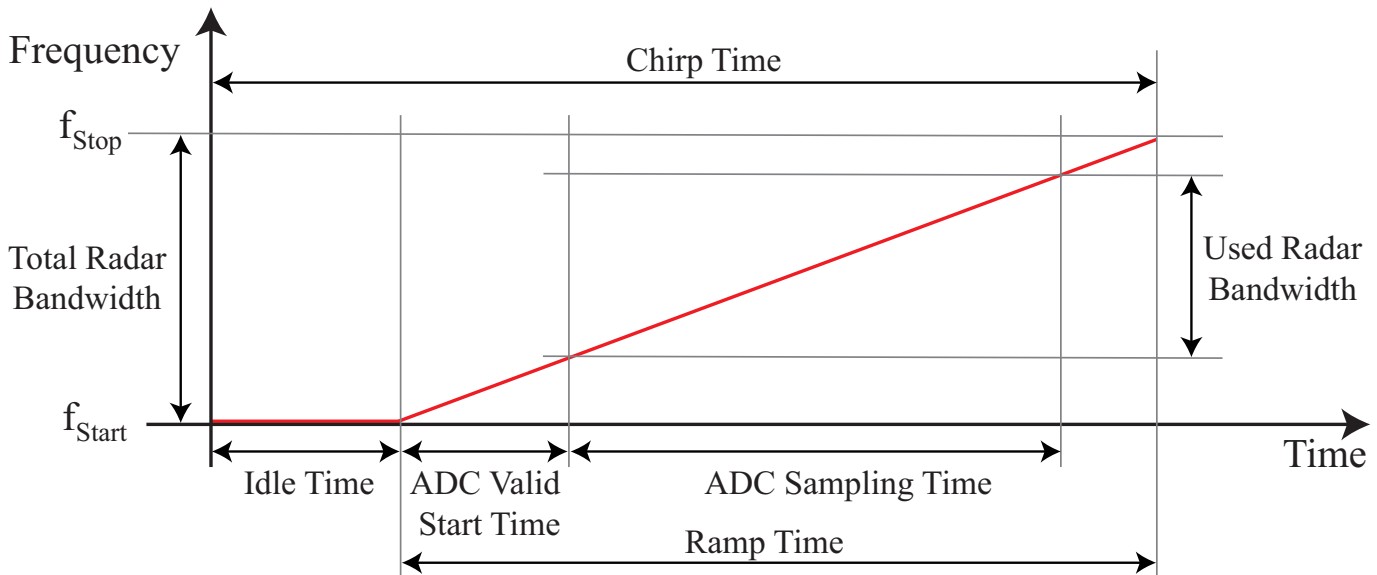

**Figure 5.** Chirp time graphical representation.

Frames are used to organize chirps transmission. Each frame can contain at least 1 chirp up to a maximum of 255 ones. Figure 6 shows how chirps are transmitted in time.

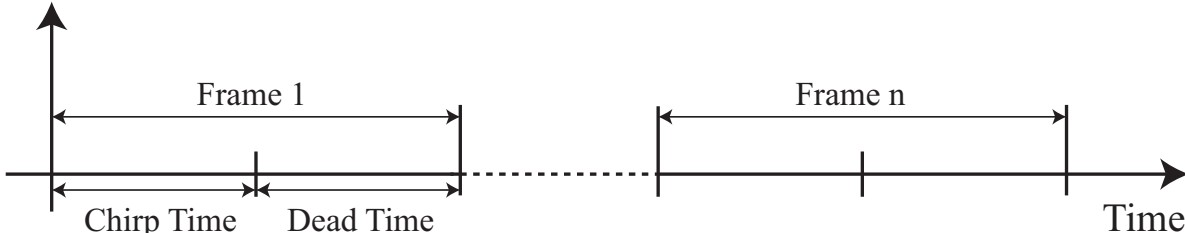

**Figure 6.** Chirps transmission within frames.

In Figure 6, the case of a single chirp per frame is illustrated. In the first part of the frame, the chirp is transmitted. In the second part, named *dead time*, the sensor neither receives nor transmits. By using this operating mode, it is possible to maximize the range of the detectable vibrational frequencies, according to Equation (15).

$$f_{max} = \frac{1}{t_{periodicity}} \tag{15}$$

where $f_{max}$ is the maximum observable vibrational frequency and $t_{periodicity}$ is the frame repetition time. The time $t_{periodicity}$ represent also the sampling time of the vibration signal.

Table 2 shows the Radar parameters configuration, which is selected in order to easily identify the specific object chosen in our experiments to be the Radar target, i.e., a wooden aluminium-coated panel, among other objects located nearby in the lab. For this purpose a resolution of 0.067 m is used, a value that is comparable to the panel thickness.

**Table 2.** Radar Parameters.

| Parameter | Value |
| --- | --- |
| Idle Time | 100 µs |
| ADC Valid Start Time | 6 µs |
| ADC Sampling Time | 63 µs |
| Used Radar Bandwidth | 3.99 GHz |
| $t_{periodicity}$ | 976 µs |

From the configuration parameters values and from Equation (2), we can state that the maximum detection distance is 4.151 m, so this is the maximum distance at which the target panel can be located from the Radar sensor. In this case, only one chirp per frame is transmitted, so the pulse repetition time is $t_{periodicity}$.

### 2.2. Radar Signal Processing

In Section 1 the capability of a Radar system to detect small vibrations is described, but the technique used for the extraction of the information can be different from technology to technology. In the case of an automotive Radar, like the one used in this work, the vibration signal can be extracted from the phase of the Beat signal [47]. Typically, an FMCW transmitted signal is composed of up-chirps and down-chirps but, as described in the previous paragraph, the tested system can transmit only up-chirps, so the signal model used in this section refers only to this configuration. The transmitted signal for a single chirp can be written as:

$$s_T(t) = exp\left[j\left(2\pi f_0 t + \pi \frac{B}{T} t^2 + \phi_0\right)\right] \tag{16}$$

where $f_0$ is the starting ramp frequency, $t$ is the so-called fast-time, $\phi_0$ is the initial phase, $B$ the bandwidth, and $T$ the chirp time. Under the model described above, it is possible to write the reflected signal as:

$$s_R(t) = \rho \cdot exp\left[j(2\pi f_0(t - \tau) + \pi \frac{B}{T}(t - \tau)^2 + \phi_0)\right] \tag{17}$$

where $\tau$ is the delay of the received signal, which is related to the distance by the relation:

$$\tau = \frac{2}{c} \cdot R(t) = \frac{2}{c} \cdot [R_0 + x(t)] \tag{18}$$

As known, in an FMCW Radar the transmitted signal is multiplied by the received signal. So, based on Equations (16) and (17), the Beat signal can be obtained from:

$$s_b(t) \approx \rho \cdot exp\left[j\left(2\pi \frac{B}{T}\tau t + 2\pi f_0 \tau\right)\right] \tag{19}$$

In Equation (20), $\tau^2$ is neglected because too smaller with respect $\tau t$, and also $x(t)$ can be assumed constant inside a chirp, so the variation can be seen only along with different chirps (slow-time dimension). As also described in [47], it is possible to write the Beat singal in terms of the slow-time and the fast-time:

$$s_b(iT + t) \approx \rho \cdot exp\left[j\left(\frac{4\pi \frac{B}{T}R_0}{c}t + \frac{4\pi f_0 R_0}{c} + \left(\frac{4\pi \frac{B}{T}t}{c} + \frac{4\pi f_0}{c}\right)x(iT)\right)\right] \approx \rho \cdot exp\left[j(2\pi f_b t + \Psi_i)\right] \tag{20}$$

Inside the previous equation, $f_b$ is the Beat frequency related to the main distance of the target $R_0$, and $\Psi_i$ depends on the vibration signal. Starting from a static target that only vibrates, its vibrations produce the effect of phase modulating the Radar signal. For this reason, the Radar ability to detect a remote static object's vibration relies on the possibility to analyze the received signal's phase variations in time. We can express the relationship between target motion and phase signal as follows:

$$\Psi_i = \frac{4\pi R_0}{\lambda_0} + \frac{4\pi}{\lambda_c} \cdot x(iT) \tag{21}$$

where:

- $R_0$ is target distance from the Radar in meters;
- $x(iT)$ is the target displacement in time;
- $\lambda_0$ is the initial chirp wavelength;
- $\lambda_c$ is the central chirp wavelength.

The minimum value of $x(t)$ able to produce a $\Psi_i(t)$ phase variation is:

$$\Delta_\Psi = \frac{\lambda_c}{4\pi}\Delta_x \tag{22}$$

where:

- $\Delta_\Psi$ is the minimum detectable phase;
- $\Delta_x$ is the minimum target displacement able to generate a phase variation.

From Equation (22) we can assert that the detection resolution is strictly bounded to the transmitted signal characteristics, and that it is possible to detect sub-millimeters vibrations even by using signals having a wavelength in the mm range.

In a real application scenario, the Beat signal is made up of many terms originating from the target (signal of interest), and from all the other objects that are in the Radar's range of sight. The Beat signal can be seen as a summation of terms representing the vibrational state of all the objects detected in the scene. It is subsequently mandatory to isolate the relevant target's contribution. Having a MIMO capable Radar, we can use both distance and azimuth information to locate the target. This approach improves the capability of the Radar to discriminate different targets.

The Beat signal can be modeled as a sum along the range-azimuth plane, by indicating with $s_{b_{r\theta}}$ the single Beat signal coming from an element of the Range-angular direction map. It is possible to write the following relation:

$$s_b(R,\Theta) = \sum_{r=1}^{Rm} \sum_{\theta=1}^{\theta_n} s_{b_{r\theta}} \tag{23}$$

where $R_m$ is the number of range bins in the range direction and $\theta_n$ is the number of angular bins, which depend on the angular resolution and the MIMO configuration. At this point, the importance of being able to separate targets is simple to understand. The improvement in range and angular resolution can separate different Beat signals coming from different targets. If the algorithm can separate much more targets, it is possible to extract with more accuracy the vibration signal; otherwise, the sensor detects the vibrations of all the targets summed together.

The practical implementation of the algorithm starts from the raw samples coming from the ADCs (one for each receiver). These samples have the form of complex numbers vectors, which represent the Beat signal evolution over time. Data can be rearranged in an $n \times 4$ matrix, where $n$ is the number of samples collected during the acquisition. We may think of this data as being cube-shaped along three main axes:

- the Fast-Time axis: samples of a single chirp;
- the Spatial Sampling axis: samples collected from different ADCs;
- the Slow-Time axis: samples from different chirps.

The target position is determined by executing an FFT analysis in the Fast-Time and Spatial Sampling axes. If the target is not moving, we can extract a data vector from Slow-Time which corresponds to the detected position. Having determined the position of the target, we can calculate the phase information as stated in Equation (21). After unwrapping the phase signal, we extract the vibrational information by inverting the formula and removing the average value.

The processing steps used can be graphically summarized by the flowchart shown in Figure 7.

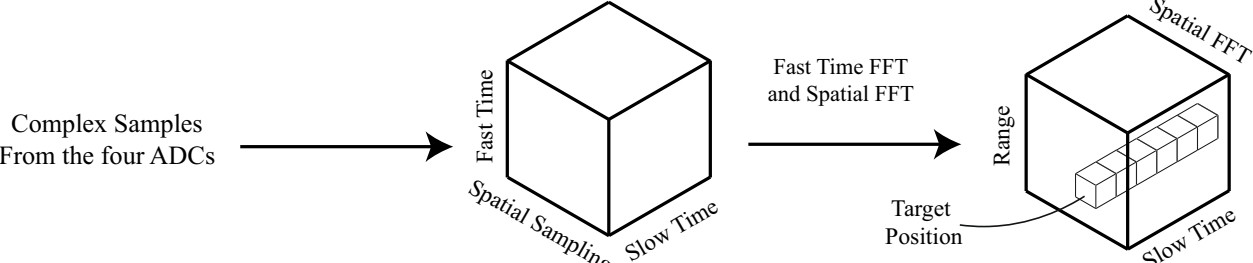

**Figure 7.** Radar signal processing flowchart.

To detect the position of the target a range-azimuth map is calculated. Since the target is not moving, its position can be identified by the same bin for all the measurements. This map is calculated only once, for identifying the position used during the processing. Figure 8 shows an example of this map. From the figure it is also possible to understand how the environment affects the algorithm capability. If only one target is present, it is simple to detect it and extract its vibration signal. In the case addressed by this work, though, the measurement area is full of objects, so being able to identify the target is more difficult, but it's a fundamental step for the correct measurement of its vibration.

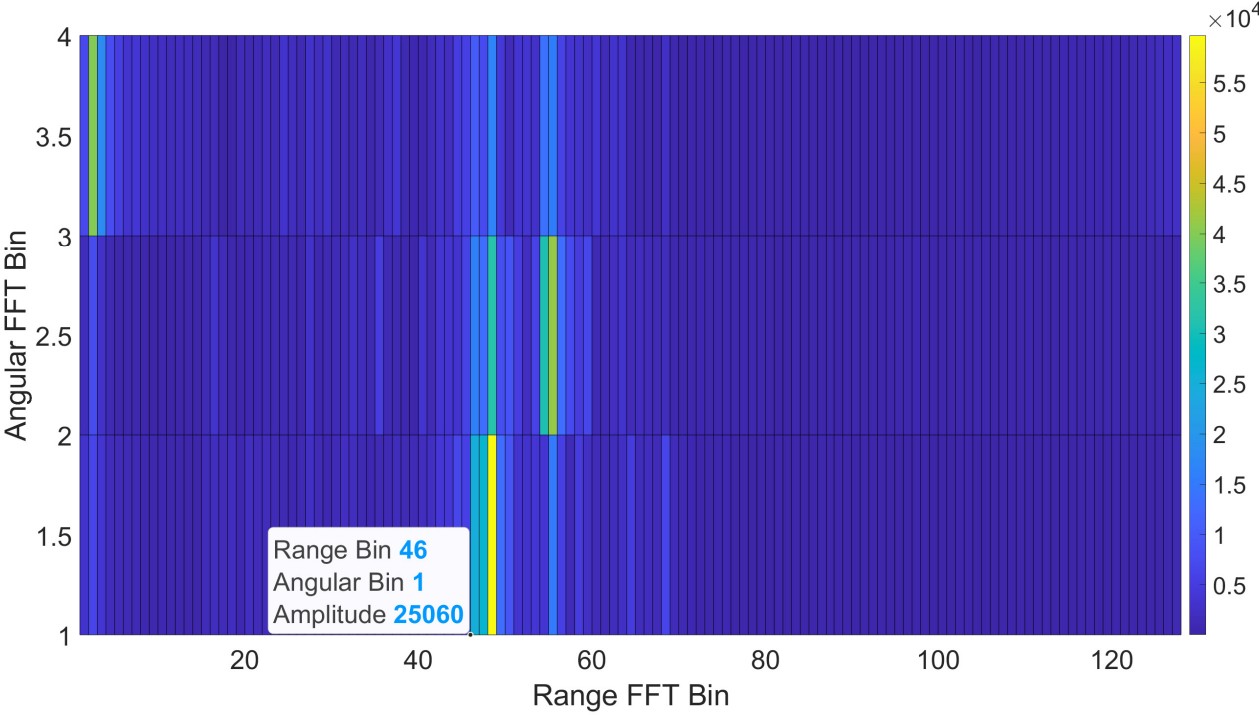

**Figure 8.** Range-angular direction map.

## 3. Experimental Tests

The area chosen for the measurements is a mechanical measurement laboratory. Inside this area, not only the measurement setup is present, but also other instruments and objects. This means that the area is not clean, but suitable to reproduce a generic environment where many objects can be present, not only the chosen target. The measurement setup includes a signal generator, a mechanical shaker, an oscillating panel (the target), and a laser vibrometer. The dimension of the panel is $46 \times 34.5$ cm and is made with a sandwich in fiberglass and the core is in polyurethane foam. The facing surface in the direction of radar and laser vibrometer is coated with aluminum tape thick of 0.5 mm. The generator is used to generate a sinusoidal signal with given stable amplitude and frequency. Signal's parameters are varied throughout the tests. The generator is connected both to an ADC (part of the laser vibrometer system), and to a mechanical shaker. The latter is bounded to an aluminium-coated panel which is the oscillating target in our setup. This setup is shown in Figure 9. Once the position of the target is chosen, the map in Figure 8 shows the angular-range information in terms of FFT bins, not the converted Range-Angle axis. The target used stands along with the angular bin no. 1 and the range bin no. 46. As it is also evident, the target structure stands along with three range bins, but the correct one is only the first, which gives the position of the vibrating panel.

In order to avoid the Radar to receive spurious reflections from the shaker support and the floor, a set of microwave absorbers have been placed on the measurement field. As mentioned above, each element that stands inside a "pixel" contributes to the beat signal, so even to the extracted vibration. For example, if we don't exclude the support of the panel, the algorithm will also detect its vibration.

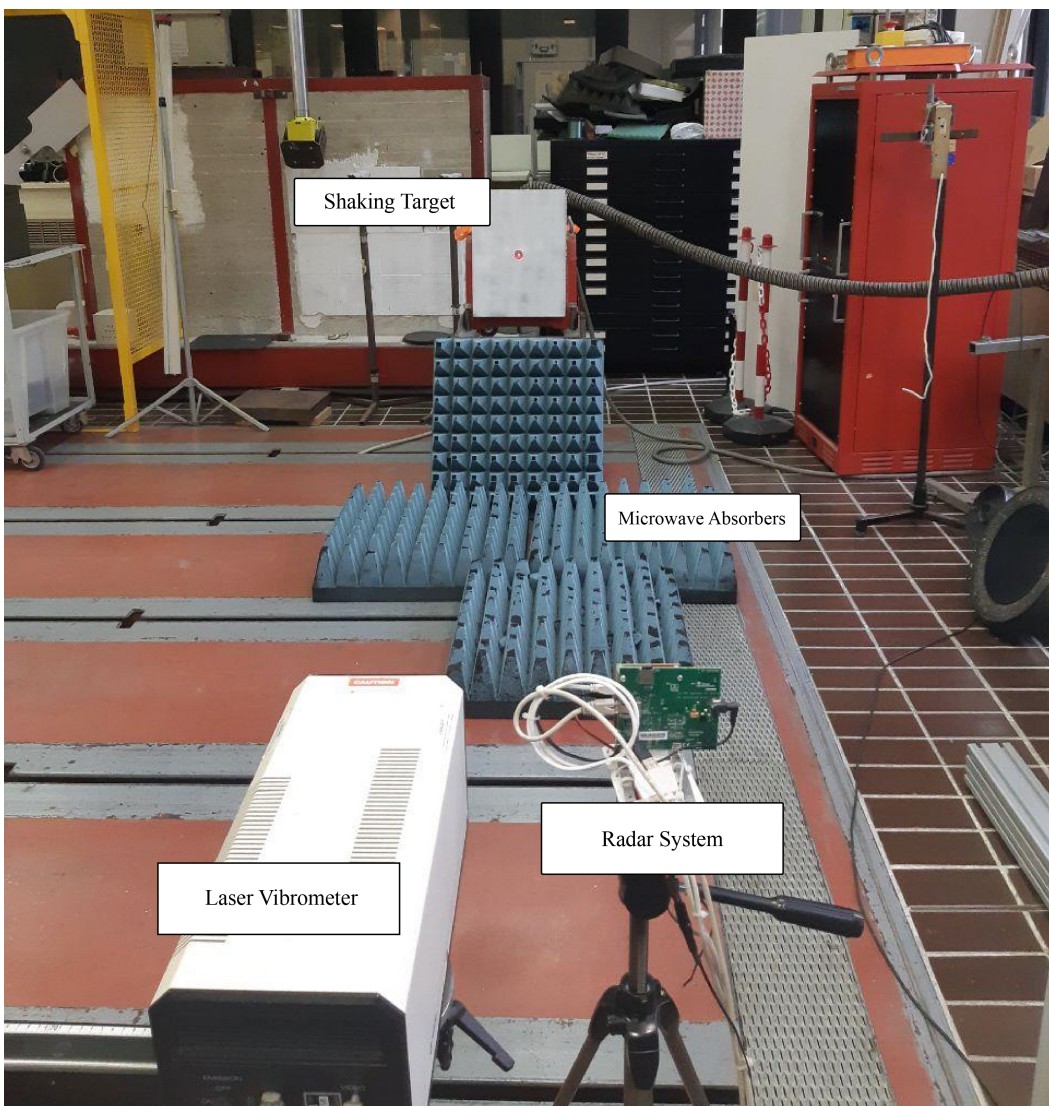

**Figure 9.** Measurement setup.

Radar and laser Vibrometer are 3.013 m far from the target and are operated simultaneously. The distance value is the range measure of the Radar. Such a distance value is taken from the Radar detection and the error is the resolution range. Both the measurement systems are aiming at the shaking panel. The laser vibrometer sight of the target is greatly improved by using the laser spot as a reference. On the other hand, the Radar aiming is obtained by orientating the PCB so that the transmitter's antenna boresight axis is lined-up with the center of the shaking panel. Two measurement campaigns have been carried out. In the former, a setup is used like the one shown in Figure 9. In the latter, the microwave absorbers are removed and the position of some objects in the Radar field of view is changed, in order to verify the repeatability of the measurement results, and their reliability. During each measurement campaign the signal generator's frequency and amplitude are chosen within a set of possible values, thus changing the target's shaking frequency and displacement accordingly. In Table 3 all these values are reported: they are chosen to produce vibration with unknown amplitude and even non-integer value of their frequency.

**Table 3.** Generator's signal amplitude (voltage) and frequency settings.

| Voltage [V] | Frequency [Hz] |
|---|---|
| 1.5, 1, 0.5, 0.25, 0.125 | 21.5 |
| 1.5, 1, 0.5, 0.25, 0.125 | 13 |
| 1.5, 1, 0.5, 0.25, 0.125 | 4.7 |

Following the system setup, measurements with both instruments used at the same time are performed, in order to determine the target vibration frequency and displacement. Each measurement lasts for 50 seconds. The Radar sensor's signals are collected and saved on a storage memory, to be processed off-line. Matlab processing will extract frequency and displacement magnitude versus time. A sample result is given in Figure 10. The number of samples collected by the Radar system is different from the laser vibrometer. For the former device, the number of samples depends on the number of transmitted measurement chirps (in this case, 51,200); for the latter, the number of samples depends on the configuration (4096). The result is evident in the harmonic analysis, where a different frequency resolution is obtained. The sampling frequency of the vibration for the Radar system is 1024 Hz, and it is 1280 Hz for the laser vibrometer. So, the frequency resolution is 0.02 Hz and 0.3125 Hz, respectively. This must be taken into account during the analysis of the results.

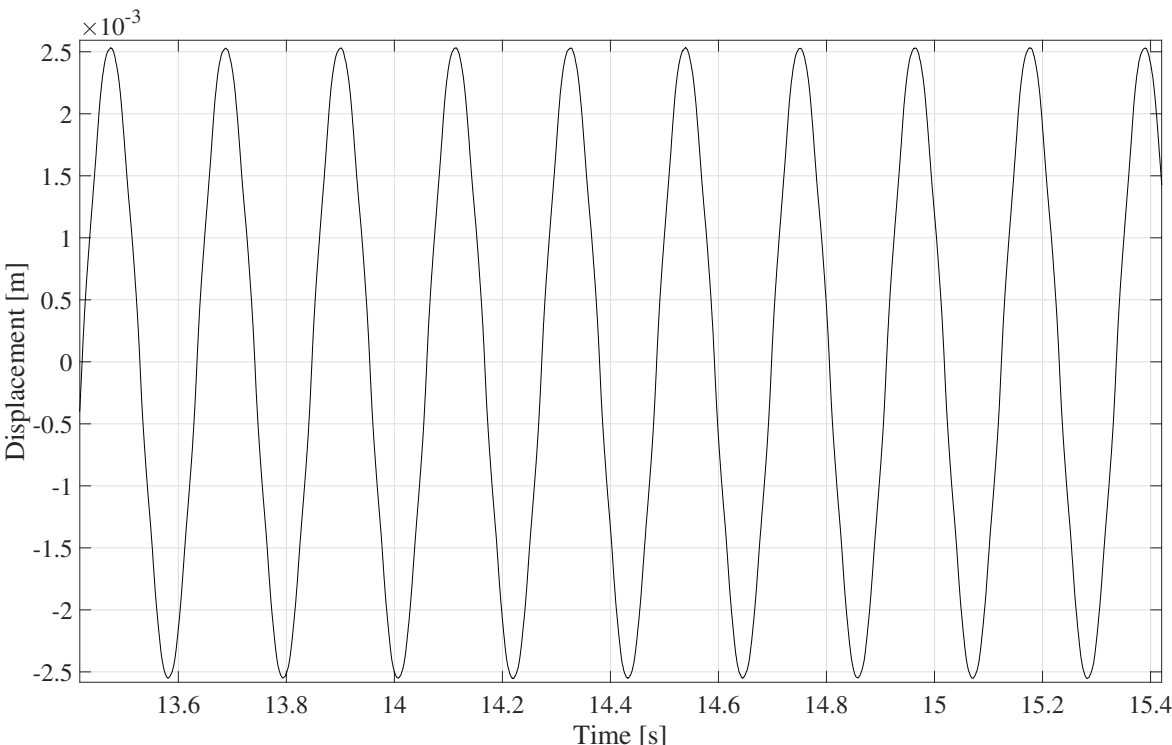

**Figure 10.** Target displacement in time, measured by the Radar.

To evaluate the Radar sensor performances in measuring the target displacement, the reference laser vibrometer is used, that can determine the target shaking displacement in real-time. The target shaking frequency as measured by the Radar equipment is directly compared with the signal generator displayed value. In Figure 11, an example of the vibrometer console output, used for the laser vibrometer measurement, is provided. The values used in experimental evaluation are taken from such a console.

For each couple of frequency and voltage settings, ten measurements are repeated in the first setup, and two in the second one. For each signal, the value of the positive relative maxima was calculated; from the average of these values, it is possible to obtain the average measure of the target displacement. This is possible because the signal generator doesn't change the generated signal along the time, so it is possible to assert that the movement of the target remains the same during all the fifty seconds of the measurement.

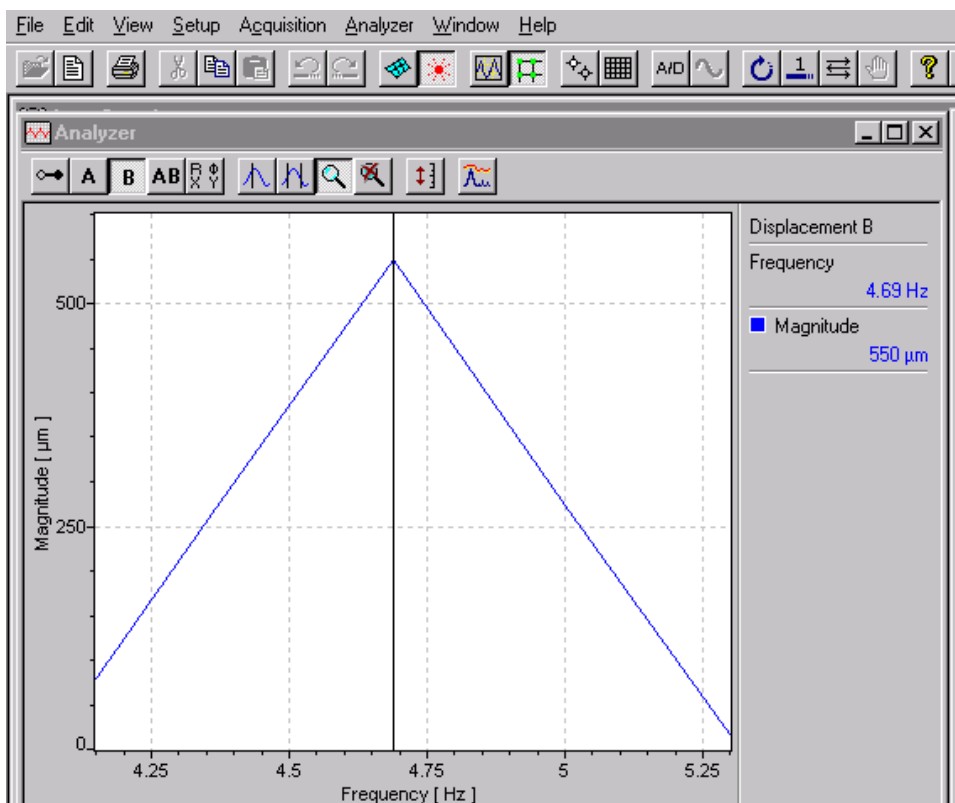

**Figure 11.** Vibrometer software display used to read the measured values.

## 4. Results

In Tables 4 and 5 the average values and standard deviations of the measurement data obtained from the vibrometer ($\overline{x}_v$) and from the Radar sensor ($\overline{x}_r$ and $\sigma_r$, as a result of signal processing) are reported.

Starting from the frequency results, these are the same at all voltage values, and for this reason Table 4 reports one value for each frequency. The laser vibrometer and the Radar system have a different resolution in frequency, but anyway it is possible to state that the Radar system produces very similar results to the vibrometer.

**Table 4.** Measurement results of the target frequency obtained in the first setup.

| Signal Gen. | Vibrometer | Radar |
|:---:|:---:|:---:|
| Frequency [Hz] | $\overline{x}_v$ [Hz] | $\overline{x}_r$ [Hz] |
| 21.5 | 21.56 | 21.49 |
| 13 | 13 | 13.13 |
| 4.7 | 4.7 | 4.69 |

The results obtained from the second setup are the same as the first one. So, the effect of other objects located inside the scene and of the ground do not degrade too much the attained measurements.

Table 5 reports the result for the detected displacement: even in this case, for the first and second setup, similar results are obtained so in the table only the former are presented.

**Table 5.** Measurement results of the average target displacement obtained in the first setup. The last column reports the difference between the Radar and the vibrometer measurement values.

| Signal Gen. | Vibrometer | Radar | | $\Delta_d$ |
|---|---|---|---|---|
| Frequency [Hz], Voltage [V] | $\bar{x}_v$ [mm] | $\bar{x}_r$ [mm] | $\sigma_r$ [mm] | [mm] |
| 21.5, 1.5 | 1.69 | 1.83 | 0.015 | 0.14 |
| 21.5, 1 | 1.08 | 1.15 | 0.004 | 0.07 |
| 21.5, 0.5 | 0.57 | 0.61 | 0.003 | 0.04 |
| 21.5, 0.25 | 0.28 | 0.32 | 0 | 0.04 |
| 21.5, 0.12 | 0.14 | 0.16 | 0 | 0.02 |
| 13, 1.5 | 2.50 | 2.88 | 0.005 | 0.38 |
| 13, 1 | 1.66 | 1.95 | 0.016 | 0.29 |
| 13, 0.5 | 0.83 | 0.92 | 0.001 | 0.09 |
| 13, 0.25 | 0.41 | 0.49 | 0.004 | 0.08 |
| 13, 0.125 | 0.20 | 0.25 | 0 | 0.05 |
| 4.7, 1.5 | 6.83 | 7.20 | 0 | 0.37 |
| 4.7, 1 | 4.62 | 4.96 | 0.012 | 0.34 |
| 4.7, 0.5 | 2.28 | 2.53 | 0.008 | 0.25 |
| 4.7, 0.25 | 1.12 | 1.16 | 0 | 0.04 |
| 4.7, 0.125 | 0.55 | 0.64 | 0.001 | 0.09 |

In the second measurement setup similar results were btained, either for displacement and frequency. By the analysis of the data, it is possible to assess that the Radar system gives stable measurement results, with a resolution of 100 μm.

The Radar used is a MIMO device, so it is possible to identify the position of the target not only in range but also in angle. This is helpful to resolve near targets and reduce the interference, especially in a radar-noisy environment. With respect to [19], the signal processing is similar but in this case the Radar is not able to discriminate targets at the same distance. This means having a degradation of the measurements. For analyzing this degradation we compare the laser vibrometer measurement with the Radar one, using the MIMO identification or not. The second Radar technique is called in this section "Radar Linear". With the setup used, the position of the target is "46" along the range FFT axis and "1" along the angular FFT axis. All the results presented are obtained from these values. In the case of the linear algorithm, only the range is present, so the identification of the position is the range bin (46). Figure 12 depicts the result of applying MIMO identification or not.

Observing Figure 12, it is possible to see how the exploitation of the MIMO capability improves significantly the quality of the measurements. The signal generated by the generator is sinusoidal, and the most similar signal is obtained when applying MIMO. Without using MIMO, the signal is still periodic but with noticeably high amplitude and phase distortion. In a measurement area where multiple objects are present and not only the designated target, it is very difficult to obtain a clean measurement of displacement and vibration frequency. In some cases, the measured displacement is completely wrong and, following the model given in Equation (23), this is the result of summing together all the contributions coming from many targets at the same distance.

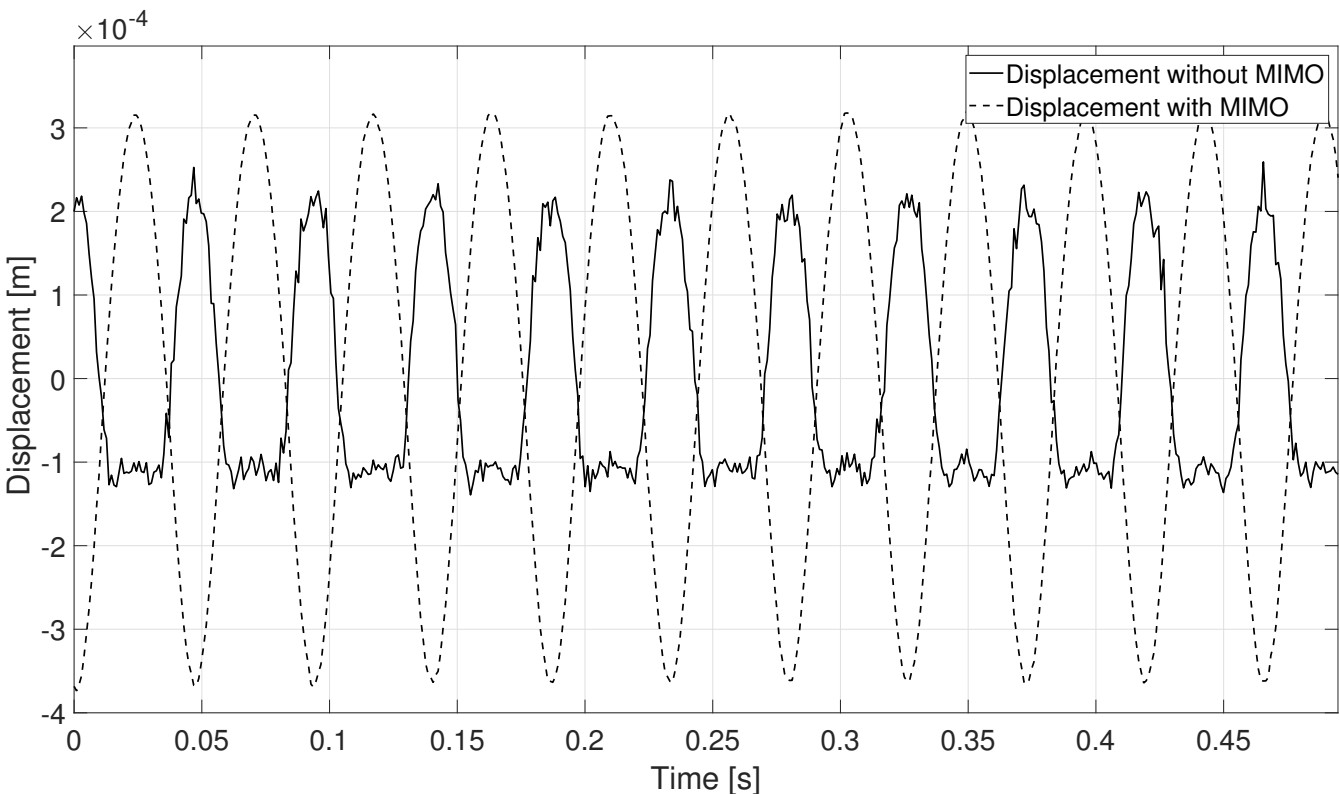

**Figure 12.** Measurement results for the test 21.5 Hz–0.25 V, when using the angular identification (enabled by MIMO) or not.

Inside Equation (23), the beat signal is the sum of multiple terms. With the application of the MIMO transmission it is possible to discriminate each component not only in range but also in angle. If the angular discrimination is not considered, the equation becomes:

$$s_b(R, \Theta) = \sum_{r=1}^{Rm} s_{b_{r\theta}}$$

(24)

Equation (24) considers all the angular components of the signal; conversely, inside Equation (23) the angular terms are not summed together so this improves the performances of the detection. The result is a beamforming of the receiver antenna diagram, which allows to separate the elements along the angle.

Another type of analysis can be focused on the harmonics of the detected vibrations. The signals' spectra are reported in Figure 13.

Comparing the results in terms of the detected vibration frequency, the laser vibrometer and the Radar system with MIMO technique achieve similar results. The scale of the spectral amplitudes shown in Figure 13 is in dB full scale [dBFS] for a better comparison. Both the systems can detect the main frequency of vibration and the other multiple harmonics with a decreasing trend. In the case of the linear Radar measurement, the detected maximum value is not always correct: in the example shown, the main frequency is around 32 Hz when the vibration is set to 4.7 Hz. The application of the algorithm with the range detection only can reach good performance when the measurement area is clear from other objects, and there is only the desired target inside the chosen range bin.

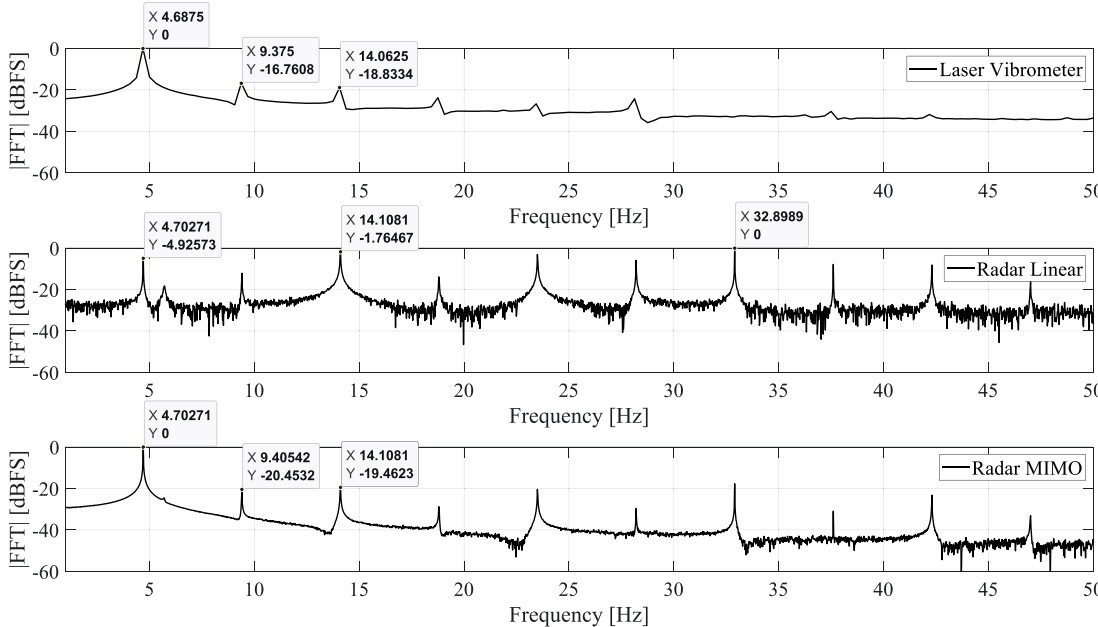

**Figure 13.** Harmonics comparison between the three measurement techniques used. The depicted spectrum is limited to 50 Hz.

## 5. Conclusions

In this work, a 77 GHz FMCW automotive Radar is used for detecting small vibrations of a target, performing frequency and displacement measurements in comparison to a laser vibrometer taken as the reference instrument. Two techniques for the Radar are analyzed, one exploiting the MIMO capability, and one without it.

From the results, it is possible to see how the Radar can detect with good performance the vibration frequency and the displacement of the target, the latter one with a resolution of hundreds of microns. Moreover, when the microwave absorbers are removed from the measurement scenario, thus increasing Radar noise and clutter, the performances remain the same. Comparing all the measurements with those provided by the vibrometer, a mean arithmetic value of $\Delta_d$ is calculated and results of 0.15 mm. This is bigger than the error stated in [47], but the setup used in this work is more realistic and the target is located at a greater distance from the sensor.

The comparison between the two Radar techniques shows how the correct detection of the position of the target is important to improve the performance of the Radar-based system. The angular detection can somehow filter out the undesired targets, and this allows to obtain a better quality Radar measurement.

**Author Contributions:** G.C., A.D.S. and P.C. designed the system; G.C., A.D.S., D.D. and P.C. performed the experimental tests; G.C. and S.S. performed the experimental analysis; G.C. and A.D.S. wrote the main part of the paper. E.G. coordinated the project, the discussion of result, and the manuscript writing. All authors have read and agreed to the published version of the manuscript.

**Funding:** This research received no external funding.

**Informed Consent Statement:** Informed consent was obtained from all subjects involved in the study.

**Data Availability Statement:** The data presented in this study are available on request from the corresponding author. The data are not publicly available as they are collected to validate the system presented and at the moment is not possible to share them.

**Conflicts of Interest:** The authors declare no conflict of interest.

## Abbreviations

The following abbreviations are used in this manuscript:

| | |
|---|---|
| FMCW | Frequency Modulated Continuous Wave |
| MIMO | Multiple In Multiple Out |
| ADC | analog-to-digital converter |
| FPGA | Field Programmable Gate Array |
| RF | Radio Frequency |
| DSP | Digital Signal Processor |
| LVDS | Low Voltage Differential Signaling |
| FFT | Fast Fourier Transform |
| ADAS | Advanced Driver Assistance System |
| SAR | Synthetic Aperture Radar |
| PCB | Printed Circuit Board |

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
