# Peer review of "Performance Evaluation of Vibrational Measurements through mmWave Automotive Radars†"

_remotesensing, doi:10.3390/rs13010098_

Round 1

Reviewer 1 Report

The paper is clearly written and brings well known problem of vibration measurements in focus again.

The algorithm is well presented.

I have a few minor questions/comments:

  1. It should be mentioned in the paper what is the size of shaking target object, and what material is it made from.
  2. Also, it would be interesting to conduct experiments for different sizes or materials, but it is not mandatory.
  3. In the introduction section authors mention that radar sensors are non expensive, which is true, especially compared to laser sensors. 

    But, authors mention later that they have used additional hardware, FPGA board which is not non expensive, for accessing the data. Also all the processing is done in Matlab, probably on some powerful PC platform.

    I would like to see the analysis of the complexity of the algorithm and the estimation of hardware implementation possibilities. Is it possible to expect that that algorithm could be implemented in the radar chip itself - DSP or Cortex part?

Author Response

We are very grateful to this Reviewer for her/his positive judgment about our work and her/his useful suggestions on how to improve the contents and presentation. In the following we give a summary of her/his concerns (noted by C)), followed by our point-by-point replies (noted by R)). The main changes in the manuscript are marked in red so that they are easily visible to the Editors and Reviewers.

C) It should be mentioned in the paper what is the size of shaking target object, and what material is it made from.

R) We add a better description of the shaking target; the dimension of the panel and the material is now present in the paper.

C) Also, it would be interesting to conduct experiments for different sizes or materials, but it is not mandatory.

R) Unfortunately at the moment it is not possible to perform other new measurements, but this is an interesting comment to consider for future developments.

c) In the introduction section authors mention that radar sensors are non-expensive, which is true, especially compared to laser sensors. But, authors mention later that they have used additional hardware, FPGA board which is not non expensive, for accessing the data. Also all the processing is done in Matlab, probably on some powerful PC platform.

I would like to see the analysis of the complexity of the algorithm and the estimation of hardware implementation possibilities. Is it possible to expect that that algorithm could be implemented in the radar chip itself - DSP or Cortex part?

R) Our experiments were developed using an evaluation board of the Texas Instruments Radar. By using only the Radar evaluation board the samples of the beat signals cannot be extracted, so the FPGA board is required for data managing and communication to a PC, running a proper Matlab software for data processing, mainly based on FFT and filtering. The complexity of such algorithms are, as known, related to amount of the elaborated data. An overall implementation of the detection system on a single board requires the design of a new board that integrates the radar sensor (with its antennas), the FPGA for data processing and the interface for data communication. This design is similar, under several points of view, to a car radar sensor, the cost is which really low.

Reviewer 2 Report

This paper proves that a 77 GHz FMCW radar can provide accurate displacement information of a vibrating object.   However, CW or pulse radars can also show similar performance, and the theoretical basis was well known. Please add a paragraph to emphasize the novelty and significant performance.   Moreover, in Fig. 12, please introduce the theory and procedures of the MIMO technique which significantly improves the accuracy.

Author Response

We are very grateful to this Reviewer for the time spent reading our paper and the constructive comments provided to improve its quality. In the following we give a summary of her/his concerns (noted by C)), followed by our point-by-point replies (noted by R)). The main changes in the manuscript are marked in red so that they are easily visible to the Editors and Reviewers.

C) This paper proves that a 77 GHz FMCW radar can provide accurate displacement information of a vibrating object. However, CW or pulse radars can also show similar performance, and the theoretical basis was well known. Please add a paragraph to emphasize the novelty and significant performance.  

R) We added a new paragraph in the introduction section with the comparison between FMCW radars and CW radars.

C) Moreover, in Fig. 12, please introduce the theory and procedures of the MIMO technique which significantly improves the accuracy.

R) The explanation of how the MIMO application improves the quality of the measurement is added.

Reviewer 3 Report

The paper is well structured, in fact, it gives the necessary information to understand its content and the description of the experimental setup is very detailed.

I suggest enlarging the Introduction section with recently published research papers in the field.

Equations formatting must be improved (i.e., parenthesis must be long enough to contain the independent variables).

Author Response

We are very grateful to this Reviewer for having emphasized, in his/her comments, the relevant points of our study and her/his useful and having provided constructive suggestions for improving the quality of the paper. In the following we give a summary of her/his concerns (noted by C)), followed by our point-by-point replies (noted by R)). The main changes in the manuscript are marked in red so that they are easily visible to the Editors and Reviewers.

C) I suggest enlarging the Introduction section with recently published research papers in the field.

R) We extended the “Related works” section with more recent papers in the field.

C) Equations formatting must be improved (i.e., parenthesis must be long enough to contain the independent variables).

R) The format of the equations has been checked and improved as suggested.

Round 2

Reviewer 2 Report

N/A